# Mobility VLA: Multimodal Instruction Navigation with Long-Context VLMs and Topological Graphs

**Hao-Tien Lewis Chiang**[*], **Zhuo Xu**[*], **Zipeng Fu**[*], **Mithun George Jacob**[†], **Tingnan Zhang**[†],
**Tsang-Wei Edward Lee**[†], **Wenhao Yu**[†], **Connor Schenck, David Rendleman, Dhruv Shah,**
**Fei Xia, Jasmine Hsu, Jonathan Hoech, Pete Florence, Sean Kirmani, Sumeet Singh,**
**Vikas Sindhwani, Carolina Parada**[‡], **Chelsea Finn**[‡], **Peng Xu**[‡], **Sergey Levine**[‡], **Jie Tan**[‡]
Google DeepMind

**Abstract:** An elusive goal in navigation research is to build an intelligent agent that can understand multimodal instructions including natural language and image, and perform useful navigation. To achieve this, we study a widely useful category of navigation tasks we call Multimodal Instruction Navigation with demonstration Tours (MINT), in which the environment prior is provided through a previously recorded demonstration video. Recent advances in Vision Language Models (VLMs) have shown a promising path in achieving this goal as it demonstrates capabilities in perceiving and reasoning about multimodal inputs. To solve MINT, we present Mobility VLA, a hierarchical Vision-Language-Action (VLA) navigation policy that combines the environment understanding and common sense reasoning power of long-context VLMs and a robust low-level navigation policy based on topological graphs. The high-level policy consists of a long-context VLM that takes the demonstration tour video and the multimodal user instruction as input to find the goal frame in the tour video. Next, a low-level policy uses the goal frame and an offline constructed topological graph to generate robot actions at every timestep. We evaluated Mobility VLA in a 836m$^2$ real world environment and show that Mobility VLA has a high end-to-end success rates on previously unsolved multimodal instructions such as "Where should I return this?" while holding a plastic bin. A video demonstrating Mobility VLA can be found here: youtu.be/-Tof__Q8_5s

**Keywords:** vision-language navigation, multimodal foundation models, long-context reasoning

## 1 Introduction

Robot navigation has come a long way. Early work relied on users specifying physical coordinates in pre-mapped environments [1, 2, 3, 4, 5, 6, 7]. Object goal and Vision Language navigation (ObjNav and VLN) [8, 9, 10, 11, 12, 13, 14] are a giant leap forward in robot usability as they allow the use of open-vocabulary language to define navigation goals, such as "Go to the couch". To make robots truly useful and ubiquitous in our daily lives, we propose another leap forward by lifting ObjNav and VLN's natural language space onto the multimodal space, meaning that the robot can accept natural language and/or image instructions simultaneously. For example, a person unfamiliar with the building can ask "Where should I return this?" while holding a plastic bin (Figure 1, upper left), and the robot guides the user to the shelf for returning the box based on verbal and visual context. We call this category of navigation tasks Multimodal Instruction Navigation (MIN).

MIN is a broad task consisting of environment exploration and instruction guided navigation. However, in many scenarios one can bypass exploration by leveraging a *demonstration tour video* that fully traverses the environment. The demonstration tour has several benefits: 1) It is easy to collect: users can teleoperate the robot or simply record a video on a smartphone while walking in the environment. There also exists exploration algorithms [9, 14] that can be used to create the tour. 2) It

---

[*]Co-first authors. [†] Core contributors. [‡] Advising leads. Corresponding authors: {`lewispro, zhuoxu,`
`jietan`}`@google.com`. ZF completed his part of work as a student researcher at Google DeepMind.

8th Conference on Robot Learning (CoRL 2024), Munich, Germany.

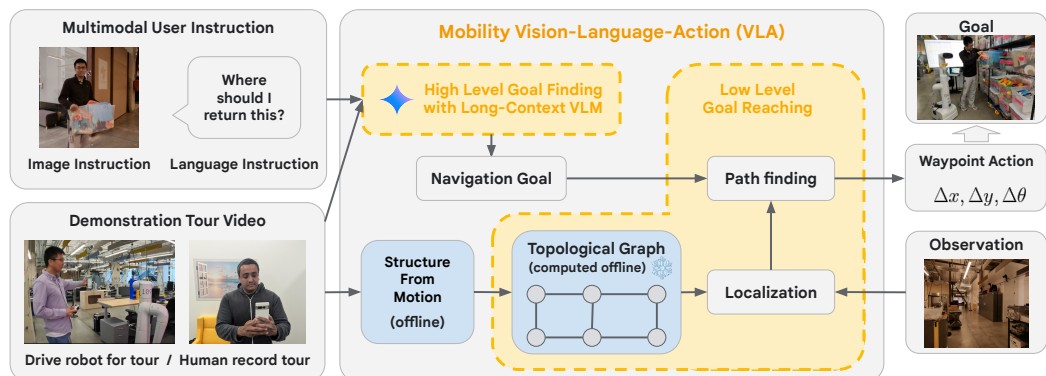

**Figure 1:** Mobility VLA architecture. The multimodal user instruction and a demonstration tour video of the environment are used by a long-context VLM (high-level policy) to identify the goal frame in the video. The low-level policy then uses the goal frame and an *offline* generated topological map (from the tour video using structure-from-motion) to compute a robot action at every timestep.

aligns with user common practice: when a user gets a new home robot, it is natural for them to show the robot around in their home, and they can verbally introduce locations of interest during the tour. 3) In certain circumstances, restricting the robot's motion in a pre-defined zone is desirable due to safety and privacy purposes [15]. To that end, in this paper, we introduce and study this category of tasks called Multimodal Instruction Navigation with Tours (MINT), which leverages demonstration tours and focuses on fulfilling multimodal user instructions.

Recently, large Vision-Language Models (VLMs) [16, 17, 18] have shown great potential in solving MINT thanks to their impressive capabilities in language and image understanding and common-sense reasoning [19], all critical pieces to achieve MINT. However, VLMs alone struggle to solve MINT due to: 1) The number of input images for many VLMs are highly limited due to context-length limitation. This severely limits the fidelity of environment understanding in large environments. 2) Solving MINT requires computing robot actions. Queries to solicit such robot actions are typically out-of-distribution from what VLMs are (pre)trained with. As a result, the zero-shot navigation performance is often unsatisfactory (we show this in Section 5.3).

To solve MINT, we present Mobility VLA, a hierarchical Vision-Language-Action (VLA) navigation policy that combines the environment understanding and common sense reasoning power of *long-context* VLMs and a robust low-level navigation policy based on topological graphs. Specifically, the high-level VLM uses the demonstration tour video and the multimodal user instruction to find the goal frame in the tour video. Next, a classical low-level policy uses the goal frame and a topological graph (constructed offline from tour frames) to generate robot actions (waypoints) at every timestep. The use of long-context VLMs addressed the environment understanding fidelity problem, and the topological graph bridges the gap between VLM's training distribution and the robot actions required to solve MINT.

We evaluated Mobility VLA in a real world ($836m^2$) office and a home-like environment. Mobility VLA achieved 86% and 90% success rates (26% and 60% higher than baseline methods) on previously infeasible MINT tasks involving complex reasoning (e.g., "I want to store something out of sight from the public eye. Where should I go?") and multimodal user instructions. We also demonstrated a leap forward in how easily users can interact with the robot, where a user takes a narrated video walkthrough in a home environment *with a smartphone* and then asks "Where did I leave my coaster?"

Our contributions include: 1) proposed a new paradigm of robot navigation: MIN and its variant MINT, which make robots significantly more helpful and intuitive to use. 2) proposed Mobility VLA as a solution of MINT, which combines long-context VLMs and topological maps. This method has significantly improved the naturalness of human-robot interaction, and greatly increased the robot usability.

## 2 Related Work

**Object and Visual Navigation** Object and image goal navigation techniques [20, 21, 22] utilize rich input modalities. These include object categories [23, 24, 25, 11], natural language instructions [26, 27], dialogue [28], goal image conditions [29, 30], and even multimodal inputs combining language and images [31]. Most of these approaches involve an active exploration phase because the robot operates without prior knowledge of the environment. Our work distinguishes itself by leveraging environment priors provided in the form of a previously collected video tour. In this regard, our work shares similarities with [14], where semantic information is obtained from past explorations, and [32, 33, 34] which utilizes memory to improve mapping and planning. However, a key difference lies in the absence of explicit semantic scene representation graphs [35, 36] in our approach, thanks to the capabilities of VLMs to process raw videos.

**Vision-language models** Prior to the emergence of large VLMs, researchers typically needed to pretrain their own visual representations for navigation tasks [37, 38, 39, 40], although some leveraged existing pretrained multimodal embeddings [31, 14]. Recent breakthroughs in large language models (LLMs) [41, 42] and VLMs [17, 18], trained on web-scale data, have paved the way for zero or few-shot navigation capabilities. This potential has been explored in various studies [43, 44, 45, 46], showcasing the diverse applications of LLMs and VLMs in navigation. These models have demonstrated the ability to: Provide navigation preferences, e.g., "stay close to marked pavements" [44]; Construct high-level motion plans, e.g., "move past the hallway towards the bedroom" [45]; Substitute object detectors, i.e., recognizing landmarks [46]; In some cases directly output trajectories [47]. Our work is most similar to [45] in the sense that a large VLM (Gemini Pro 1.5 [17]) is used to generate high-level navigation plans for the robot, but differs from the the previous work in that our VLM directly outputs the navigation goal for the low-level controller to consume and generate navigation commands.

## 3 MINT Problem Formulation

The MINT task considered in this paper takes as input a demonstration tour video and a multimodal user instruction. The robot must navigate to certain goal location(s) to satisfy the user's instruction.

Under this setting, the demonstration tour video consists of a sequence of first-person view image frames $F = \{f_i | f_i \in \mathbb{R}^{H \times W \times 3}, i = 1, 2, ..., k\}$ taken during a tour of the environment, where $k$ is the number of frames in the video. In addition, optional natural language narratives can be added to certain frames $N = \{n_j | n_j \in \texttt{str}, j \in [1, 2, ..., k]\}$. The multimodal user instruction can be just a text instruction $d \in \texttt{str}$ (e.g., "Where can I find a ladder?"), or both text and image instructions $I \in \mathbb{R}^{H \times W \times 3}$ (e.g., "Where can I get something to clean this?" + The robot sees the user pointing to a dirty whiteboard).

We aim to produce a navigation policy $\pi(a|O, F, N, d, I)$, where $O \in \mathbb{R}^{H \times W \times 3}$ is the robot's current camera observation. The policy emits an embodiment-agnostic waypoint action $a \in \mathbb{R}^3$ representing longitudinal translation ($\Delta x$), lateral translation ($\Delta y$), and rotation along the vertical axis ($\Delta \theta$), all in the robot-centric frame. We assume that the robot has an embodiment-specific mechanism to execute waypoint actions.

## 4 Mobility VLA

Mobility VLA is a hierarchical navigation policy (Figure 1) with online and offline components. In the offline phase, a topological graph $G$ was generated from the demonstration tour $(N, F)$. Online, the high-level policy takes the demonstration tour and the multimodal user instruction $(d, I)$ to find the navigation goal frame index $g$, which is an integer corresponding to a specific frame of the tour. Next, the lower-level policy utilize the topological graph, the current camera observation ($O$) and $g$ to produce a waypoint action ($a$) for the robot to execute at each timestep.

$$g = h(F, N, d, I) \tag{1}$$
$$\pi(a|O, F, N, d, I) = l(a|G, O, g) \tag{2}$$

where $h$ and $l$ are the high and low-level policies.

## 4.1 Demonstration Tour and Offline Topological Graph Generation

Mobility VLA utilizes a demonstration tour of the environment to solve MINT. This tour can be given by a human user via teleoperation, or by simply recording a video on a *smartphone* while walking in the environment.

Mobility VLA then constructs a topological graph $G = (V, E)$ offline, where each vertex $v_i \in V$ corresponds to the frame $f_i$ from the demonstration tour video $(F, N)$. We use COLMAP [48, 49], an off-the-shelf structure-from-motion pipeline to determine the approximate 6-Degree-of-Freedom camera pose for each frame and store it in the vertex (see Section 7.1 for details). Next, a directed edge is added to $G$ if the target vertex is "in front of" the source vertex (less than 90 degrees away from source vertex's pose) and is within 2m.

Compared to traditional navigation pipelines (e.g., map the environment, identify traversable areas and then construct a PRM [50]), the topological graph approach significantly simpler as it captures the general connectivity of the environment based on the tour trajectory.

## 4.2 High-Level Goal Finding with Long-Context Multimodal VLMs

During online execution, the high-level policy leverages the common sense reasoning ability of VLMs to identify a navigation goal from the demonstration tour that satisfies a wide range of multimodal, colloquial and often ambiguous user instructions. To this end, we prepare a prompt $P(F, N, d, I)$ consisting of interleaving text and images. A concrete example of $P$ for the multimodal user instruction "Where should I return this?" in Table 1 is shown below:

```
You are a robot operating in a building and your task is to respond to the user
command about going to a specific location by finding the closest frame in the
tour video to navigate to.
These frames are from the tour of the building last year.
[Frame 1 Image f₁]
Frame 1. [Frame narrative n₁]
...
[Frame k Image fₖ]
Frame k. [Frame narrative nₖ]
This image is what you see now. You may or may not see the user in this image.
[Image Instruction I]
The user says: Where should I return this?
How would you respond? Can you find the closest frame?
```

The VLM returns an integer goal frame index $g$.

## 4.3 Low-level Goal Reaching using Topological Graphs

Once the goal frame index $g$ is identified by the high-level policy, the low-level policy (Algorithm 1) takes over and produces a waypoint action at every timestep (Eq. 1).

---

**Algorithm 1 Low-level Goal Reaching Policy**

---

1: **Input:** goal frame index $g$, offline-constructed topological graph $G$.
2:
3: **while** timestep $\leq$ maximum steps **do**
4:     Get new camera observation image $O$
5:     Get start vertex $v_s$ and robot pose $T$ by localizing $O$ in $G$
6:     **if** $v_s == v_g$ **then**
7:         Navigation goal reached, break
8:     **end if**
9:     Compute $S = [v_s, v_1, ..., v_g]$, the shortest path between $v_s$ and $v_g$.
10:     Compute waypoint action $a$ from the relative pose between $T$ and $v_1$
11:     Execute $a$ on robot
12: **end while**

---

At every timestep, we use a real-time hierarchical visual localization system (described briefly below, please see Section 7.1 for more details) to estimate the pose of the robot $T$ and the closest start

vertex $v_s \in G$ (line 5) using the current camera observation $O$. This localization system finds k-nearest candidate frames in $G$ w.r.t a global descriptor [51], and then computes $T$ through PnP [52]. Next, the shortest path $S$ on the topological graph between $v_s$ and the goal vertex $v_g$ (the vertex corresponding to $g$) is identified by Dijkstra's algorithm (line 9). Finally, the low-level policy returns a waypoint action which is simply the $\Delta x, \Delta y, \Delta \theta$ of the next vertex $v_1$ in $S$ relative to $T$ (line 10).

# 5  Experiments

To demonstrate the performance of Mobility VLA and gain further insights into key designs, we design experiments to answer the following research questions (RQs):

**RQ1:** Does Mobility VLA perform well in MINT in the real world?

**RQ2:** Does Mobility VLA outperform alternatives thanks to the use of long-context VLM?

**RQ3:** Is the topological graph necessary? Can VLMs produce actions directly?

| Reasoning-Free (RF) | | Small Object (SO) | | |
| --- | --- | --- | --- | --- |
| Text instr. ($d$) | Goal frame ($g$) | Text instr. ($d$) | | Goal frame ($g$) |
| Take me to a conference room with a double door. | | Where can I borrow a hand sanitizer? | | |
| **Reasoning-Required (RR)** | | **Multimodal (MM)** | | |
| Text instr. ($d$) | Goal frame ($g$) | Text instr. ($d$) | Image instr. ($I$) | Goal frame ($g$) |
| I want to store something out of sight from public eyes. Where should I go? | | Where should I return this? | | |

**Table 1:** Examples of user instructions in 4 categories (better in color).

We highlight the key experimental setup below and leave details to Section 7.

**Environments.** We evaluate Mobility VLA for MINT in an real office environment occupied by humans (Figure 2a). It is 836m$^2$ and cluttered with everyday items such as shelves, desks and chairs.

**Robot.** We use a wheel-based mobile manipulator (Figure 2b) to evaluate Mobility VLA. The robot uses a MPC-based algorithm [53] to execute the waypoint action ($\Delta x, \Delta y, \Delta \theta$ in the robot-centric frame) while avoiding obstacles.

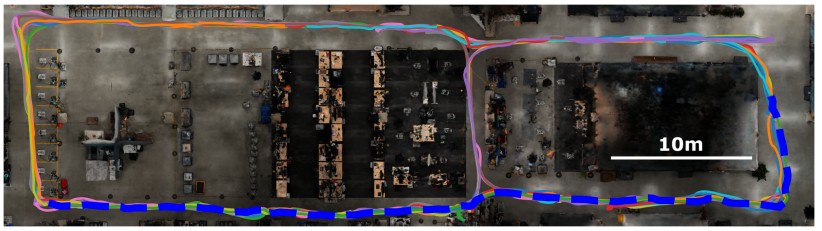 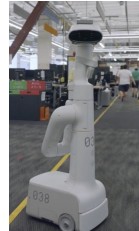

**(a)** Top down view of the 836m$^2$ office environment with all paths taken during experiments. The longest robot path (60.22m) is in dashed blue.

**(b)** Our mobile manipulator

**Figure 2:** Experiment setup.

**Demonstration Tour.** We collect the demonstration tour by teleoperating the robot with a gamepad. All corridors are traversed twice from opposite directions. The resulting tour is roughly 16 minutes

long (948 frames @ 1Hz) and we add narratives during the tour "Temp desk for everyone" and "Lewis' desk" to frame 5:28 and 7:14 respectively to enable personalized navigation.

**Multimodal User Instructions.** We crowd-sourced 57 user instructions in 4 categories. This includes: 20 Reasoning-Free (RF), 15 Reasoning-Required (RR), 12 Small Objects (SO), and 10 Multimodal (MM) instructions (Examples are in Table 1, full list in Section 7.7). Importantly, "Reasoning Required" instructions do not mention the specific object or location the robot needs to navigate to, and the destination of Multimodal instructions are nearly impossible to infer without the image modality in the instruction. As far as we know, prior works were not designed for or evaluated against these two categories of tasks, and they are the key differentiator between MINT and ObjNav and VLN.

## 5.1 RQ1: Mobility VLA's robust high end-to-end performance in the wild

To evaluate Mobility VLA in MINT in the real world, we randomly select 5 user instructions per category and evaluate Mobility VLA's performance from 4 random starting poses (location and yaw) that are at least 20 m away. We use Gemini 1.5 Pro [17] as our long-context multimodal VLM.

**High end-to-end success rate.** Table 2 shows that Mobility VLA has a high end-to-end navigation success rate in most user instructions categories, including previously infeasible Reasoning-Required and Multimodal instructions. However, the success rate is significantly lower in the Small Object category. This is not unexpected given the limited tour video resolution. Mobility VLA also has a reasonable SPL (Success Rate weighted Path Length), indicating that the topological graph does not incur a high path length penalty. Lastly, Mobility VLA successfully incorporated the personalization narratives in the demonstration tour. It correctly navigated to different locations when responding to essentially the same instructions, but from different users (moved to frame 7:14 when asked "I'm Lewis, take me to a temp desk please." and moved to frame 5:28 when asked "Hi robot, I'm visiting, can you take me to a temp desk?"). See the supplementary video for examples.

|  | Reasoning-Free | Reasoning-Required | Small Objects | Multimodal |
|---|---|---|---|---|
| Goal Finding SR | 80% | 80% | 40% | 85% |
| Goal Reaching SR | 100% | 100% | 100% | 100% |
| End-to-end SR | 80% | 80% | 40% | 85% |
| SPL | 0.59 | 0.69 | 0.38 | 0.64 |

**Table 2:** Mobility VLA end-to-end navigation Success Rate (SR) and SPL of various user instruction types in the *real* Office environment.

**Robust low-level goal reaching.** Table 2 also shows the robustness of Mobility VLA's low-level goal reaching policy (100% success rate) in the real world, with the demonstration tour recorded months prior to experiments when many objects, furniture, and lighting conditions had been different.

**Large-scale sim confirms high end-to-end success rate.** To further investigate the end-to-end performance, we leverage simulations to scale evaluation numbers. Concretely, we created a high fidelity simulation reconstruction of the office environment using NeRF [54] (see Section 7.7 for details and example images), and evaluate Mobility VLA against 20 language instructed tasks with 50 random starting poses per task. Our experiment resulted in 90% high level goal finding and 100% low level goal reaching success rates, with a total of 900 successful end-to-end execution (full results in Table 8 in Section 7.3).

**Generalization to a home-like environment using a smartphone tour.** To show Mobility VLA's generality and ease of use, we conducted a proof-of-concept experiment in a real home-like environment (details in Section 7.2). Rather than giving the robot a teleoperated tour, we use a *Google Pixel 6 smartphone to record the demonstration tour*, and then evaluated Mobility VLA end-to-end with 4 Reasoning-Required and 1 Small Object user instructions with 4 random starts each. The success rate is 100% with a SPL of 0.87. This shows that 1) Mobility VLA performs well regardless of environments, and 2) it is extremely easy to deploy, as the user can simply use their smartphone to record a tour of their environment, upload to the robot and then immediately start giving instruc-

tions. To our knowledge, this level of generality and ease of use had never been shown in robot navigation.

## 5.2 RQ2: Long-context VLM outperforms alternatives on high level goal finding

We investigate how well alternative methods perform compared to Mobility VLA to answer whether using long-context multimodal VLMs is the key to solve MINT. Concretely, we compare the following baselines:

> **CLIP-based retrieval**: We reproduce the high-level goal finding module of NLMap [14] by adopting OWL-ViT [55] for region proposal and CLIP [56] for sub-regions and full-images embeddings extraction for tour frames. We then perform goal frame retrieval using CLIP embeddings of the instruction language and image. State-of-the-art work like ESC [22], ZSON [31], and CLIP-on-Wheels [57] also fall into the category of this baseline.

> **Text-Only Mobility VLA**: Similar to [58], where the multimodal demonstration tour is captioned by a VLM frame-by-frame to form a "text tour". An LLM (Gemini 1.5 Pro [16]) then uses the text tour to produce the goal frame index.

**Mobility VLA outperforms comparisons.** Table 3 shows that high-level goal finding success rates of Mobility VLA are significantly higher than comparison methods. Given the 100% low-level success rate, this high-level goal finding success rates are representative of end-to-end success rates.

| Success Rates | Reasoning-Free | Reasoning Required | Small Objects | Multimodal |
|---|---|---|---|---|
| CLIP-based retrieval | 35% | 33% | 25% | 20% |
| Text Only Mobility VLA | 70% | 60% | **50%** | 30% |
| Mobility VLA (Ours) | **95%** | **86%** | 42% | **90%** |

**Table 3:** High-level goal finding Success Rates of Mobility VLA compared to baselines

**Processing high frame rate tour videos with long-context VLMs is critical for success.** Feeding a full demonstration tour of a large environment into non-long-context VLMs is challenging since each image requires hundreds-of-token budgets. One solution for reducing input tokens number is *reducing* the tour video frame rate, at the cost of intermediate frames loss. Table 4 shows that the high-level goal finding success rate decreases as the tour frame rate decreases. This is unsurprising since a lower frame rate tour can sometimes miss the navigation target frame. In addition, comparing state of the art VLMs, only Gemini 1.5 Pro yields satisfactory success rate thanks to its long 1M token context-length.

| Frame | GPT-4V [18] | | | | GPT-4o [59] | | | | Gemini 1.5 Pro [17] | | | |
|---|---|---|---|---|---|---|---|---|---|---|---|---|
| Rate | RF | RR | SO | MM | RF | RR | SO | MM | RF | RR | SO | MM |
| 0.2 FPS | 60% | 53% | 17% | 30% | 75% | 40% | 25% | 50% | 95% | 67% | 36% | 60% |
| 1 FPS | Exceeds token limit | | | | Exceeds token limit | | | | **95%** | **86%** | **42%** | **90%** |

**Table 4:** High-level goal finding Success Rates with regards to various user instruction types (presented in the order of Reasoning Free (RF), Reasoning Required (RR), Small Objects (SO), MultiModal (MM)) as a function of VLM models (column) and multimodal demonstration tour Frames Per Second (FPS) (row). All VLMs were queried in June 2024.

We also showcase one selected qualitative comparison example for high-level goal finding of all candidates approaches in Figure 3. When given the multimodal instruction of "I want more of this." and a picture of several Coke cans on a desk, Mobility VLA correctly identified the frame containing the refrigerator which it should lead the user to. On the other hand, CLIP-based retrieval finds a region in which a water bottle and some stuff are on a desk to be most similar to the full instruction image, given it is hard to extract "what the user want" from the instruction image using Owl-ViT. GPT-4o incorrectly attempts to find the frame closest to the instruction image, while GPT-4V refuses

to give a frame number since it was unable to find a frame where beverages are. Lastly, the Text only approach cannot understand whether "this" refers to the Coke cans or the office setting, since it relies only on caption of the instruction image. The full LLM responses can be found in Section 7.5

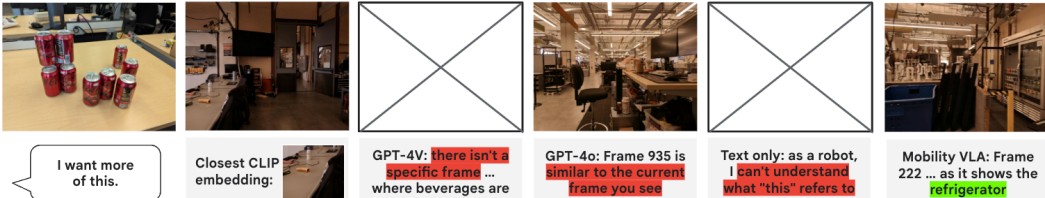

**Figure 3:** Qualitative comparison of Mobility VLA and other approaches on a multimodal instruction. The bottom row shows the intermediate output of each approach.

Altogether, experiments in this section show that the *long-context multimodal VLM capability is critical in solving MINT*. In addition, at present, only the Gemini 1.5 Pro VLM provides satisfactory success rate due to its long 1M token context-length.

### 5.3   RQ3: Topological graphs are critical for success

Mobility VLA uses a hierarchical architecture to harness long-context VLM's reasoning capability and uses a topological graph to produce waypoint actions. Is this necessary? Can we prompt the VLM to output waypoint actions directly?

**Topological graphs are critical for navigation success.** Table 5 shows the end-to-end performance of Mobility VLA in simulation compared to prompting the VLM to output waypoint actions directly (prompt and details in Section 7.6). The 0% end-to-end success rate shows that Gemini 1.5 Pro is incapable of navigating the robot zero-shot w/o the topological graph. Empirically, we found that Gemini almost always outputs the "move forward" waypoint action regardless of the current camera observation. In addition, the current Gemini 1.5 API requires the upload of all 948 tour images at every inference call, resulting in a prohibitively expensive 26s per-step running time for the robot to move just 1m. On the other hand, Mobility VLA's high-level VLM spends 10-30s to find a goal index and then the robot navigates to the goal using the low-level topological graph results in a highly robust and efficient (0.19s per step) system for solving MINT.

|  | Direct Waypoint Action Output | Goal Index Output + Topological Graph |
|---|---|---|
| Success Rate | 0% | **90%** |
| SPL | - | 0.84 |
| Per-step inference Time | 25.90±8.36s | **0.19±0.047s** |

**Table 5:** End-to-end navigation results for different VLM output formats in the *simulated* Office environment.

## 6   Discussion

In this paper, we present Mobility VLA, a new paradigm of navigation policy for solving MINT. Mobility VLA achieved 86% and 90% end-to-end success rates on previously infeasible navigation tasks involving complex reasoning and multimodal user instructions in a large real world environment. We also demonstrated a leap forward in how easily users can interact with the robot, where a user records a video walkthrough in a home environment with a smartphone and then asks "Where did I leave my coaster?"

**Limitation: (1) Lack of exploration.** The current version of Mobility VLA relies on a demonstration tour, and does not explore the environment automatically. However, existing exploration mechanisms such as frontier exploration or diffusion-based exploration [39] can be easily integrated during the demonstration tour. **(2) Long VLM inference time impedes natural user interactions.** The inference time of high-level VLMs is round 10-30 seconds, resulting in users awkwardly waiting for the robot to respond. However, it is possible to cache the demonstration tour, which takes up roughly 99.9% of the input tokens, in order to significantly improve inference speed.

## Acknowledgement

We thank Anthony Brohan, Bauyrjan Jyenis, Coline Devin, Dmitry Kalashnikov, Dorsa Sadigh, Joy Ortiz, Kuang-Huei Lee, Matthew Mounteer, Nikhil J Joshi, Peter Anderson, Quan Vuong, Robert Baruch, Sergey Yaroshenko, Sherry Moore, Tianli Ding, Ying Xu, Yixin Lin and Yuheng Kuang for their support, advice, comments and discussions during the project.

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

# 7 Appendix

## 7.1 Structure-from-Motion and Hierarchical Localization

We use COLMAP [48], an off-the-shelf structure-from-motion pipeline to estimate the pose of the robot for each frame in the tour (i.e. reference images), 3D point landmarks in the environment (see Figure 4) and their corresponding 2D projections across all reference images (i.e. 2D-3D correspondences).

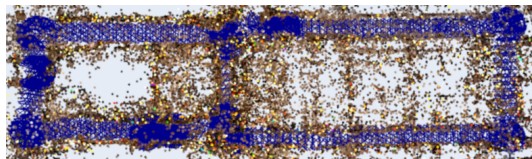

**Figure 4:** Top-down view of the COLMAP result of the office environment: 3D point landmarks and reference image poses (blue).

The poses are used to build a fully connected topological graph. The tour frames $F$, 3D landmarks and 2D features are used in our own implementation of a real-time hierarchical localizer. The method is hierarchical since it divides localization of the observed image $O$ into two steps: a global search to determine a set of candidate reference images close to $O$ followed by local feature matching and pose estimation.

In the global search, the candidate set $C \subseteq F$ of k-nearest (w.r.t. the $l^2$-norm of a global image descriptor [51]) tour frames to $O$ is determined. 2D features [60] in $O$ are matched to the 2D features of each frame in $C$. Using the pre-computed 2D-3D correspondences, we establish correspondences between 2D features in $O$ and 3D landmarks observed in the tour.

Given the set of 2D-3D correspondences for each frame in $C$, the pose of $O$ is computed by solving the corresponding Perspective-n-Point problem [52]. The pose with the most inlier 2D-3D correspondences is selected as $T_O$.

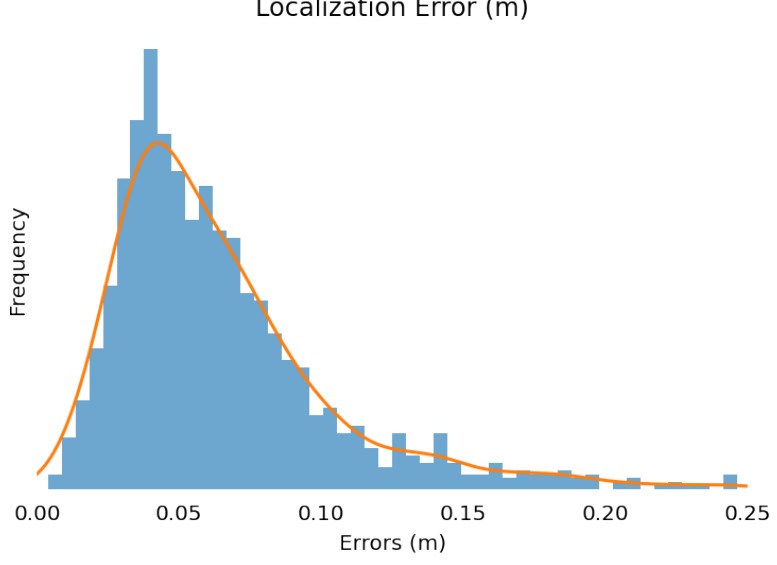

**Figure 5:** Localization error: median ATE = 0.056m.

When $T_O$ is used to determine the closest vertex on $G$, the scale-ambiguity characteristic of monocular structure-from-motion systems is inconsequential to the high-level goal-finding policy. However, when computing the waypoint action for low-level navigation (see Algorithm 1), the scale factor is utilized to generate metrically accurate actions.

We evaluated localization accuracy by comparing it to groundtruth computed by localizing unseen test images with COLMAP in a slow but accurate offline process (see Figure 5). Note that the method failed when $O$ was blurry or feature-sparse and the system was forced to fall back to the last known pose. Since we would eventually receive a feature-rich, non-blurry frame, this limitation did not affect end-to-end performance.

## 7.2    Home-Like Environment Experiment Setup

A handheld Pixel 6 smartphone tour of a home-like environment was collected (see Figure 6). The tour is 75 seconds long and contains 224 frames (3 Hz).

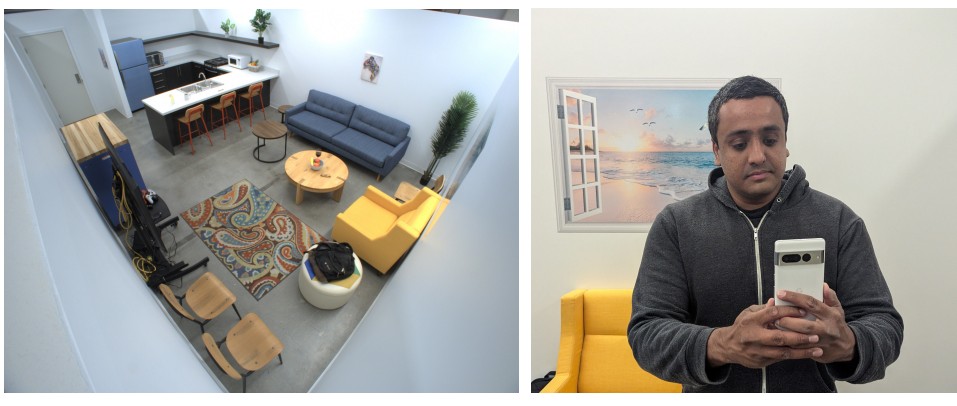

(a) Home-like Environment                      (b) Smartphone Tour

**Figure 6:** Collection of the handheld smartphone tour (right) of a home-like environment (left).

Once the topological graph was built from the tour, Mobility VLA was evaluated end-to-end with 5 instructions (see Table 6) and 4 random start points.

| Instruction | Category |
|---|---|
| Where did I leave my coaster? | Small Objects |
| I want to heat up some food. Where should I go? | |
| Where can I keep this ice cream cold? | Reasoning-required |
| Where can I go number 2? | |
| Where can I eat my dinner? | |

**Table 6:** Instructions in the home-like environment

Even though the images from the camera are significantly different (see Figure 7) from the robot's camera with a collection trajectory independent of robot motion, we achieved 100% success rate with an SPL of 0.87.

## 7.3    Additional Experiments

We also investigate if strictly multimodal user instructions (instructions that are nearly impossible to answer without the image) can be answered by the text modality alone. To this end, we replace the image part of the multimodal user instructions with its caption. Table 7 shows the high-level goal reaching success rate of such setup in the Text Instruction columns compared to feeding VLMs the image (MM Instruction column).

**Multimodal user instructions requires multimodal demo tour and image instructions.** Table 7 shows that the success rate is much higher when multimodal demo tour and image instructions are fed to the VLM (lower right corner). Replacing the image with its caption significantly reduces success rate.

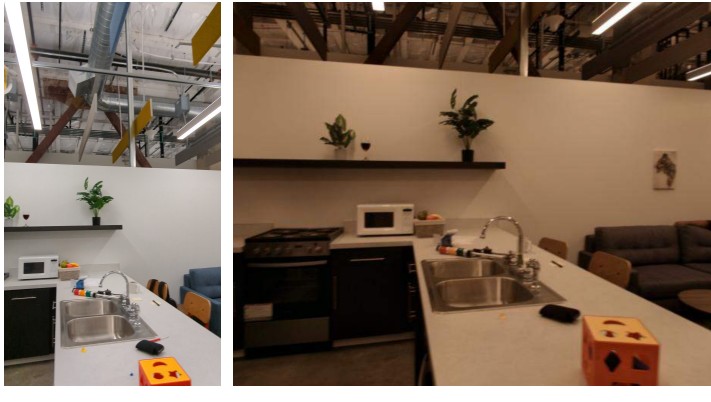

(a) Smartphone        (b) Robot

**Figure 7:** Images from the smartphone (left) and robot (right) cameras.

| Success Rates | GPT-4o Text Instruction | GPT-4o MM Instruction | Gemini 1.5 Pro Text Instruction | Gemini 1.5 Pro MM Instruction |
|---|---|---|---|---|
| Text Tour | 0.10 | 0.10 | 0.20 | 0.20 |
| Multimodal Tour | Exceeds token limit | Exceeds token limit | 0.40 | **0.90 (Ours)** |

**Table 7:** High-level goal finding Success Rates of multimodal user instructions as a function of VLM models and instruction representations (columns) and tour modalities (row). In MM Instructions columns, the robot's current camera observation is fed directly into the VLMs. In Text Instructions columns, the camera observation is captioned by Gemini 1.5 Pro and the caption text is then fed into the VLMs. The text tour was captioned w/ Gemini 1.5 Pro

## 7.4 Large-Scale end-to-end sim details

To further investigate Mobility VLA's end-to-end performance, we use simulations to conduct large-scale evaluations. To that end, we created a high fidelity reconstruction of the Office environment in simulation using NeRF [54].

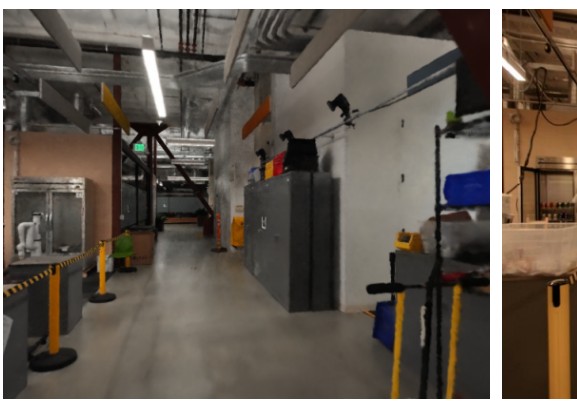 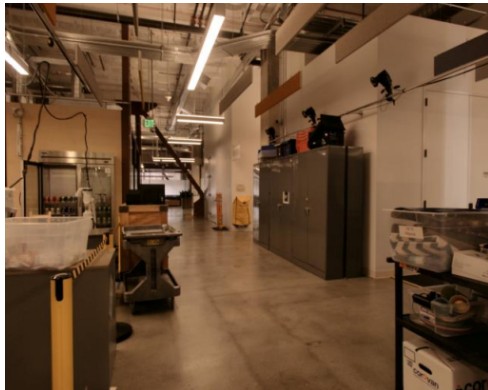

(a) NeRF render of the simulated Office environment      (b) Real image of the Office environment

**Figure 8:** Side-by-side comparison of the NeRF rendering and real image.

The Office environment was reconstructed in simulation using ZipNeRF [61]. To accomplish this, we capture 3,244 images of the office environment using a Sony A7 IV camera with a Rokinon 12mm wide-angle lens. The capture is taken uniformly along the walkways of the office facing in all directions. COLMAP [48] is then used to determine the image poses and camera intrinsics. We train a NEural Radiance Field (NeRF) using the technique described in ZipNeRF [61]. This NeRF is used to derive the assets needed for simulation. To generate the collision mesh, we distill

a mesh from the NeRF using the technique described in [62]. We then use the distilled mesh as a reference to construct the collision mesh by hand from mesh primitives using Blender. To render realistic camera images, we align the NeRF with the simulated environment and render using the same camera intrinsics as the real robot's cameras. This ensures that the image distributions of the simulator and the real environment match as closely as possible. Figure 8 shows a side-by-side comparison of a simulated render using NeRF and an image from the robot's camera at the same location in the real environment.

We then evaluate Mobility VLA in simulation by randomly picking 10 instructions in both Reasoning-Free and Reasoning-Required categories each with 50 random starting poses. Unfortunately we cannot directly user instructions in the real Office environment as the NeRF reconstruction was done a few months prior from when the demonstration tour in real was collected. We omit the Small Objects and Multimodal instructions due to limited visual fidelity and challenges in including humans in NeRF

|  | Reasoning-Free | Reasoning Required |
| --- | --- | --- |
| High-Level Goal Finding SR | 90% | 90% |
| Low-Level Goal Reaching SR | 100% | 100% |
| End-to-end SR | 90% | 90% |
| SPL | 0.83 | 0.84 |

**Table 8:** End-to-end navigation Success Rate (SR) and SPL of various user instruction types in the *simulated* Office environment.

## 7.5 Full VLM intermediate output in Figure 3

| GPT-4V | GPT-4o | Text Only Mobility VLA | Mobility VLA |
|---|---|---|---|
| It appears you are interested in finding more Coca-Cola Zero Sugar cans, specifically the ones with the Marvel character design. To assist you in locating more of these cans within the building, I would need to navigate to the area where beverages or pantry supplies are stored. Based on the images provided, there isn't a specific frame that shows a location where beverages are stored or a pantry area. However, common places to look for additional cans of soda in an office or building environment would be a kitchen, break room, or storage area where supplies are kept. If you can provide more context or direct me to a specific area within the building where beverages are typically stored, I can attempt to find the closest frame that matches that description. Otherwise, I recommend checking common areas where refreshments are usually available. | To find more of the Coca-Cola cans, you should navigate to the closest frame in the tour video that shows the area where the cans are likely located. The current frame you see is Frame 945, which shows a workspace area. The closest frame in the tour video to this location is Frame 935. This frame shows a workspace area with desks and equipment, which is similar to the current frame you see. Navigate to Frame 935 to find more of the Coca-Cola cans. | This is a tricky one! As a robot, I can't really understand what "this" refers to in your request. It could be the soda cans, the office setting, or something else entirely. To help me understand, could you please be more specific? For example, you could say:* "I want to see more pictures of soda cans."* "I want to see more pictures of offices." * "I want to see more pictures of whatever is in this image". "Once I understand what you're looking for, I can try to find the closest matching frame from my database". | Frame 222 is the closest frame as it shows the same refrigerator |

**Table 9:** Full VLM intermediate output wrt to the multimodal user instruction shown in Figure 3

## 7.6 Prompt for direct VLM waypoint actions output

```
You are a robot operating in a building and your task is to respond to the user
command about going to a specific location by finding the closest frame in the
tour video to navigate to.
These frames are from the tour of the building last year.
[Frame 1 Image f_1]
Frame 1. [Frame narrative n_1]
...
[Frame k Image f_k]
```

```
Frame k. [Frame narrative n_k]
This image is what you see now. You may or may not see the user in this image.
[Image Instruction I]
The user says: Where should I return this?
Could you select and answer the most appropriate action to take now among
'left', 'forward' and 'right', which correspond to respectively? Answer:
```

## 7.7 User Instructions tested in the Office environment

**Reasoning-Free Instructions.**

1. Can you take me to the building map?
2. Where can I find a ladder?
3. Take me to the exit
4. Where can I find some paper cups?
5. gray trash can.
6. Take me to robot number 109.
7. Take me to a blue area.
8. I want to borrow my friend's scooter, can you take me to it?
9. Take me to a conference room with a double door.
10. I need a tripod, where can I find it in this office?
11. Take me to a whiteboard.
12. Where are the gray cabinets again?
13. I heard there's a cool dark-backgrounded poster, where is it?
14. where can I find a long wooden bench?
15. Take me to a two-paned door
16. I'm Lewis, take me to a temp desk please.
17. Hi robot, I'm visiting, can you take me to a temp desk?
18. Take me to a white shelf
19. Take me to a plant
20. where can I find a moving box?

**Reasoning-Required Instructions.**

1. There is a fire, where should I find tools to fight the fire?
2. I'm thirsty.
3. I'm here to water things, please guide take me to them.
4. Help me dispose of this cardboard box.
5. Take me to a room with a closed door.
6. I want to store something out of sight from public eyes. Where should I go?
7. I left my drink on a cart, can you take me to it?
8. Can you take me upstairs?
9. I need to charge my phone, please help.
10. I heard there is a place to see lots of robots?
11. I need to sit down.
12. Can you take me somewhere to lie down?
13. Where can I find something cold?
14. I'm tired. Where can I rest?
15. I want to draw something.

**Small Objects Instructions.**

1. Where is the Jackery portable power station?
2. where is the bench with a bag of chips on it?
3. where can I find a fire extinguisher?
4. Where can I borrow a hand sanitizer?
5. I heard there is a cute tiny traffic cone, where is it?
6. I need a xbox controller.
7. Did you see my white water bottle?
8. Where can I find a fire alarm switch?
9. Can you help me find my cat mask?
10. My friend told me to get his moving box under his desk, can you help me find it?
11. take me to the tombstone I heard so much about.
12. where can I find a toy cart?

**Multimodal Instructions.** See Table 10 below.

| # | Text | Image |
|---|------|-------|
| 1 | Where should I return this? |  |
| 2 | I don't want this anymore. Can you help me? |  |
| 3 | Where can I charge this? |  |
| 4 | I want more of this. |  |
| 5 | Can you follow the directions on the whiteboard? |  |
| 6 | I don't think this is supposed to be here, where should it go? |  |
| 7 | Where can I get something to clean this? |  |
| 8 | Where can I use this? |  |
| 9 | Where can I see the most amount of this? |  |
| 10 | Can you take me somewhere to NOT see this? |  |

Table 10: Multimodal Instructions.

## 7.8 A colab for reproducing the goal reaching part of Mobility VLA

```
Copyright 2024 Google LLC.
SPDX-License-Identifier: Apache-2.0
```

Prerequisites:

```
!pip install pytubefix
```

Obtain a tour from YouTube:

```python
from pytubefix import YouTube
import cv2
from PIL import Image

def download_youtube_video(url, output_path="video.mp4"):
  """Downloads a YouTube video to the specified output path.

  Args:
    url: The URL of the YouTube video.
    output_path: The path where the video should be saved.
  """
  yt = YouTube(url)
  stream = yt.streams.get_by_itag(244)  # This is a 480p stream.
  stream.download(filename=output_path)

# Replace with your YouTube video URL
video_url = "https://www.youtube.com/watch?v=C_jSIKC1OyY"
download_youtube_video(video_url, output_path="video.mp4")
```

```python
from moviepy.editor import VideoFileClip
from PIL import Image
import tqdm

def mp4_to_pil_images(mp4_file, save_every_n_frames=30, img_downsample_ratio=2):
  """Converts an MP4 file to a list of PIL Images using MoviePy.

  Args:
    mp4_file: Path to the MP4 file.

  Returns:
    A list of PIL Images.
  """

  clip = VideoFileClip(mp4_file)
  images = []
  frame_counter = 0
  for frame in tqdm.tqdm(clip.iter_frames()):
    if frame_counter % save_every_n_frames != 0:
      frame_counter += 1
      continue
    # Convert NumPy array to PIL Image
    pil_image = Image.fromarray(
      frame[::img_downsample_ratio, ::img_downsample_ratio, ...]
    )
    images.append(pil_image)
    frame_counter += 1
  return images

images = mp4_to_pil_images("video.mp4")
print(len(images))
```

See an example image in the tour:

```
images[96]
```

Initialize Gemini 1.5 Pro:

```python
import google.generativeai as genai

# Init Gemini 1.5 Pro
main_key = "myAPIKey"  #@param

genai.configure(api_key=main_key)
gen_config = genai.types.GenerationConfig(temperature=0.0)
gemini = genai.GenerativeModel(
  model_name="gemini-1.5-pro-latest", generation_config=gen_config
```

```
)
gemini_text = genai.GenerativeModel(
  model_name="gemini-1.5-pro-latest", generation_config=gen_config
)

response = gemini.generate_content(["hi"])
print(response.text)
```

Utility functions:

```
#@title Utils
def get_gemini_response(user_input, tour_chunks):
  contents = [
      "You are a robot operating in a building and your task is to respond to the user"
      "command about going to a specific location by finding the closest frame in the tour"
      "video to navigate to. These frames are from the tour of the building last year.",
      *tour_chunks,
      "The user says: ",
      user_input,
      "How would you respond? Can you find the closest frame?",
  ]
  response_stage1 = gemini.generate_content(contents)
  try:
    stage1_text = response_stage1.text
  except:
    print(response_stage1)
    raise ValueError('Gemini Refuses to respond..')
  return stage1_text, contents

def get_goal_idx(stage1_response_text):
  # Need clarification check
  contents_clarification = [
      stage1_response_text,
      "Can you find only one unique frame number in the sentence above? Tell me that "
      "frame number and nothing else."
  ]
  idx_text = gemini_text.generate_content(contents_clarification)
  idx_text = idx_text.text
  idx = -1
  try:
    idx = int(idx_text)
  except: pass
  return idx
```

Make tour context chunks:

```
#@title Make tour context chunks
# This is where you can add custom label or nicknames to any frame.
custom_pois = {0: "my favorite plant", 150: "Lewis' desk", 192: "The kid's room"}

def get_framed_chunks_from_imgs(imgs):
  chunks = []
  for i, img in enumerate(imgs):
    frame_caption = f"Frame {i}."
    if i in custom_pois:
      frame_caption = f"Frame {i}. {custom_pois[i]}."
    chunks.append(img)
    chunks.append(frame_caption)
  return chunks

tour_chunks_custom = get_framed_chunks_from_imgs(images)
```

Goal finding wrt an user instruction:

```
#@title Goal finding wrt an user instruction
user_instruction = "Where is the office?"  #@param {type:"string"}

# stage1_text is the raw gemini response, contents is the prompt sent to Gemini
stage1_text, contents = get_gemini_response(user_instruction, tour_chunks_custom)
print(f"Gemini Raw Response:\n{stage1_text}")

idx = get_goal_idx(stage1_text)
print(f"Closest frame index:\n{idx}")
images[idx]
```

