# OpenReview forum: "Mobility VLA: Multimodal Instruction Navigation with Long-Context VLMs and Topological Graphs"
_robot-learning.org/CoRL/2024/Conference — CoRL 2024_

### Official Review · Reviewer_y2wr · 2024-07-12
**A solid experimental contribution, though with limited theoretical advance**

**Originality:** 2
**Technical Quality:** 4
**Clarity Of Presentation:** 5
**Potential Impact:** 3
**Recommendation:** 3
**Confidence:** 4

**Review:**

This is a solid experimental contribution and one that pushes the limits of what is possible with VLMs in the context of robot assistants. The paper is well written and very easy to follow. The real-world demonstrations are impressive and convincing.

I am encouraged and convinced by the general direction of the work, in which the output of the VLM is used to inform lower-level planning and decision-making. This synergy takes advantage of the benefits of both. However, it is largely unsurprising that the VLM was not well-suited for directly outputting low level motion commands to reach a frame of interest. There exist multiple works in the space of developing foundation models for local navigation that have shown the same thing, necessitating . While I still think the attempt worthwhile, it might be worth amending that section to acknowledge that there are other approaches more appropriate for this purpose: e.g., refs [43, 44] already referenced in the related work.

My only hesitation about this work is that the theoretical contribution or insight is relatively limited. It is somewhat unsurprising that VLMs are capable of attending to the most relevant images corresponding to a "common sense" or vague query and the central advance of this work is the scale of the experiments, which are mostly made possible by the use of Gemini 1.5 and its enormous context window. However, I think this is only a relatively small conceit and this work adds to the growing body of work that integrate foundation models with classical planning methods to achieve new and impressive capabilities.

Finally, there is little to no discussion of the failure modes of the system (see my comment below), so additional discussion therein would be a helpful addition to the work and contribute to the discussion on future work and possibilities for further advances.

Smaller comments:
- The authors might also want to add to the Limitations section a comment about the cost of using the VLMs. At the time of this writing, it seems that each query costs on the order of $5, which is quite a large cost for a more general agent. I do not think this is a flaw of the work, but some additional transparency would strengthen the limitations.
- There are many capitalization issues in the references: e.g., [45] "Palm 2" -> "PaLM 2" and [49] "Navgpt" -> "NavGPT". Please review the reference list and correct.

**Quality Of The Limitations Section:**

3

**Questions For Rebuttal:**

- For a system with 90% success rate and navigation via a topological graph of the environment, one might expect that the SPL be very close to 0.9, but is instead 0.84. Can the authors clarify where the other 0.06 SPL was lost or how the SPL was computed? Is it merely due to imperfect localization or navigation on the physical platform?
- What are the types of failure modes exhibited by the most powerful system (the one using Gemini 1.5 Pro)? 90% success rate is impressive, but it would be informative to understand the sorts of failures that represent the other 10%. Does the VLM return a frame in the wrong location or fail to suggest a frame at all? If it suggests incorrect frames, how far off are they typically? Are they usually nearby the intended goal or far off? Quantitative results are unnecessary for this, as I think it would be most useful to build the reader's intuition with a qualitative discussion.

**Robotics Focus:**

4

**Summary Of Paper:**

This paper presents an approach to using vision-language models (VLMs) for a task the authors call "multimodal instruction navigation with demonstration tours" (MNIT), in which a robot is given a video of an initial tour of a (large) space and an instruction (e.g., "where can I put this away" + a picture of the object of interest) and must indicate the frame in the tour video corresponding to the location the robot must travel. To achieve this aim, the authors first use the immense context length of SOTA VLMs and feed frames from throughout the tour video, ~950 for the experiments in the paper, and ask the language model to specify which frame most relevant for solving the query. Using that frame as a target coordinate, the robot then plans and follows a series of waypoints in a topological map built offline via structure from motion on the tour video. Experiments show impressive performance, as the robot achieves 90% success rate on high-level tasks.

**Summary Of Recommendation:**

Overall, I believe this to be a solid experimental contribution and a well written paper. The results are impressive and demonstration on a real robot is commendable. The theoretical contribution is a bit small, but I think the strength of the physical experiments alone are sufficient to compensate for this.

---

### Official Review · Reviewer_21ra · 2024-07-19

**Originality:** 2
**Technical Quality:** 2
**Clarity Of Presentation:** 4
**Potential Impact:** 2
**Recommendation:** 4
**Confidence:** 4

**Review:**

## Quality
The paper presents Mobility VLA, a hierarchical navigation policy integrating long-context Vision-Language Models (VLMs) and topological graphs for Multimodal Instruction Navigation with demonstration Tours (MINT). The technical methodology is sound, and the experiments are well-executed, showcasing high success rates. However, some aspects, such as the detailed workings of VLMs and topological maps, need further elaboration to fully appreciate the system's functionality.

## Clarity
While the paper is generally clear and well-organized, the contribution statement lacks technical depth and specificity. The explanation of how VLMs and topological maps are used is brief, leaving readers with an incomplete understanding of the system's mechanisms. Additionally, the experimental setup is underexplained.

## Originality
The approach of using demonstration tours as environmental priors and integrating long-context VLMs is novel. However, the paper does not sufficiently compare this approach to similar navigation problems, such as semantic mapping and navigation, which could potentially solve the same issues. Highlighting the novelty in replacing semantic mapping with video recordings needs more critical analysis.

## Significance
The significance of the work lies in its practical application, enhancing the usability of robot navigation systems by allowing multimodal instructions. The high success rates in real-world environments are promising. However, the limited scope of the experiments and the lack of comparison to alternative methods reduce the impact of these results.

## Strengths
Combining long-context VLMs with topological graphs for handling multimodal instructions is a creative and effective solution.
Using demonstration tours as environmental priors aligns with common user practices and enhances system usability.
The experiments, though limited, provide clear evidence of the system's capabilities in real-world settings.

## Weaknesses
- The paper lacks clear identification of the specific navigation problem addressed (e.g., ObjNav, VLN) and a detailed critical analysis of the novelty of replacing semantic mapping with video recordings for topological mapping and navigation planning.
- There is a need for more detailed explanations of the topological graph generation using COLMAP, its role in navigation, a comprehensive comparison with traditional semantic mapping methods, and a statistical user study to evaluate the intuitiveness of human-robot interaction and the impact of VLM context length on performance.
- The paper does not provide a thorough review and comparison of existing object search navigation methods using VLMs and target-driven navigation approaches using reinforcement learning, missing an analysis of their strengths and weaknesses relative to Mobility VLA.

While the paper presents a promising approach to multimodal instruction navigation, it needs significant revisions to address the highlighted weaknesses. The contribution statement should be more precise and technical, the experiments should be expanded, and detailed explanations of the VLM and topological map usage are necessary. Furthermore, a thorough comparison with semantic mapping methods and a critical analysis of the system's novelty are essential. Addressing these issues will significantly strengthen the paper and make it a strong candidate for publication in CoRL.

**Quality Of The Limitations Section:**

2

**Questions For Rebuttal:**

- Clarify the specific type of navigation problem addressed by Mobility VLA, such as ObjNav or VLN. - Provide a more technical and precise contribution statement.
- Expand the experiments to include a larger variety of instructions and environments. Evaluate more than five instructions per category to better demonstrate the system’s robustness and generalizability.
Provide a more detailed explanation of how VLMs are used to identify goal frames, including the creation of prompts, processing of inputs, and the decision-making process.
- Elaborate on the generation of topological graphs using structure-from-motion techniques and how they are used in the low-level navigation policy.
- Include a comparison with traditional semantic mapping and navigation methods. Discuss whether these methods could solve the MINT problem and highlight the pros and cons of using video recordings instead.
- Conduct a user study to quantify the intuitiveness of human-robot interaction with Mobility VLA compared to traditional approaches. Provide a statistical analysis of these results.
- Explain how the length of the context in VLMs impacts the performance of Mobility VLA. Provide quantitative analysis to illustrate this relationship.
- Provide a more detailed and comprehensive literature review on existing object search navigation methods using VLMs, as well as recent target-driven navigation approaches using other learning methods such as reinforcement learning. Highlight the strengths and weaknesses of these approaches in comparison to Mobility VLA. Here is an example: NavFormer: A transformer architecture for robot target-driven navigation in unknown and dynamic environments, 2024

**Robotics Focus:**

4

**Summary Of Paper:**

The paper introduces Mobility VLA, a hierarchical navigation policy for Multimodal Instruction Navigation with demonstration Tours (MINT). Mobility VLA leverages long-context Vision-Language Models (VLMs) to interpret demonstration tour videos and multimodal user instructions, identifying goal frames within the video by using interleaved text and image prompts to match the user’s request to frames from the tour. Offline-generated topological graphs, created using structure-from-motion techniques like COLMAP to establish 6-DoF camera poses for each frame, capture the environment’s connectivity based on the tour trajectory. These graphs guide the robot’s actions at each timestep by mapping the current observation to the closest vertex and computing waypoints using the shortest path to the goal. This approach improves robot usability by allowing natural language and image instructions, making navigation tasks more intuitive and effective. Evaluated in a real-world office environment, Mobility VLA demonstrated high success rates in complex navigation tasks, enhancing human-robot interaction.

**Summary Of Recommendation:**

The paper presents an innovative approach, Mobility VLA, which combines long-context Vision-Language Models (VLMs) and topological graphs for handling multimodal instruction navigation using demonstration tours.  While the technical methodology is sound and the initial results are promising, the paper requires significant revisions to address several weaknesses. These include a vague contribution statement, insufficient experimental details, lack of comprehensive comparisons with alternative methods, and a need for more in-depth explanations of the VLM and topological graph usage. Additionally, the paper should include a user study to quantify the system's intuitiveness, analyze the impact of context length in VLMs, and provide a more detailed literature review. Addressing these issues will greatly strengthen the paper and support its claims. With these improvements, the paper could become a strong candidate for publication in CoRL.

---

### Official Review · Reviewer_395i · 2024-07-28
**Interesting task but weak contribution**

**Originality:** 3
**Technical Quality:** 3
**Clarity Of Presentation:** 5
**Potential Impact:** 1
**Recommendation:** 2
**Confidence:** 4

**Review:**

Strengths: The paper is well-written and organized.  The proposed problem is original and relevant. Different set of results support the paper.

Weaknesses: The reviewer does not see the "robot learning" part of the paper. It is mostly an engineering paper assembling different pieces of code. The performance of Mobility-VLA relies mainly on Gemini-Pro 1.5, not on the author's contribution.

1. As mentioned above , the reviewer does not see any technical contribution in the "robot learning" field. The experiments results show that the peformance of Mobility-VLA depends only on Gemini-Pro.

2. The low-level navigation with the topological map is straightfforward. If the goal frame (destination) and map are known, what is the value of the low-level goal reaching policy over a classic path planning algorithm?

3. Based on 2 and 3, is the proposed problem (in the current form) challenging enough and suitable for the community?. I.e. , One would just need to wait for Google to release a better version of Gemini, to achieve better accuracy.

4. The authors overlook the narrative text in the paper. How does it differ from the instructions?  Could the authors provide examples?  Why is it needed and when?

5. Related to 4, The reviewer would have expected a more thorough study about the need of such a multimodal dataset. How is the narrative  text important to solve the task?

6. Unfortunately, the authors did not produce/release a proper dataset for such a task.

**Quality Of The Limitations Section:**

2

**Questions For Rebuttal:**

The rebuttal questions are provided above.

Review update: Based  on the interactions with the authors, the reviewer changed the recommendation to weak reject. The reviewer is convinced by the relevance of the work, but not by the contribution. For such a work showing a different use of MLLMs, the reviewer would have expected  a more thorough experimental study to compensate the lack of technical novelty.

**Robotics Focus:**

4

**Summary Of Paper:**

This paper introduces  the multimodal instruction navigation with demonstration tours. Unlike usual VLN problems, the MINT  challenge focuses on multimodal instructions (text+ images). To solve this navigation task, the authors proposed mobility VLA, combining a Multimodal large language model to predict the image frame corresponding to the instruction, and a  low-level topological navigation algorithm to generate waypoints.

**Summary Of Recommendation:**

Mainly an engineering work where the performance depend only on a specific MLLM.

---

### Author Rebuttal · Authors · 2024-08-14

We thank the reviewers and the AC for all the constructive and helpful feedbacks! We have updated the paper based on the suggestions, improving the contribution statement to reflect technical depth, enriching related works with a complete reference list, making explicit and comprehensive analysis and comparisons between the proposed task and method with those in the literature, and adding more details for methodology and experiment setting explaination such that the audience could understand better and reproduce our results. We also answered the clarifying questions that the reviewers raise, and are in process of releasing a colab with publicly-available dataset for the community to try Mobility VLA, to be associated with the camera ready version paper. The latest updated draft is attached as the rebuttal file.

We want to highlight that we believe our proposed problem and methodology are highly relevant to the robot learning field and would be beneficial for the community. While not training the long-context VLMs, Mobility VLA presents a novel use of large multimodal models that enables previously unattainable robot capability in navigation, and the novel use of publicly-available LLM and large multimodal models without additional training is common for robot learning literatures. We believe that this kind of exploratory research that tries to answer the question "what can large models do for us out of the box, so that we we can benefit from their internet-scale knowledge" is very valuable to researchers in the robot leraning community.

In terms of the future impact of this work, analyzing learned internal representations and tying it more closely with verifiable planning could be an exciting direction of future work enabled by this paper. We also observe several valuable directions to be explored by the community to improve MINT (see Response to reviewer 395i's response on August 13th for details). Finally, the experiment results the Mobility VLA paper shows could potentially inspire VLM researchers to improve VLMs for navigation and Embodied AI.

---

### Decision · Program_Chairs · 2024-09-04

**Decision:**

Accept

**Comment:**

# Strengths
1. The proposed problem is highly relevant to the field.
1. The approach to combining VLMs and topological graphs is interesting.
1. The real-world demonstrations showcase the effectiveness of the approach.
1. The paper is well-written and well-structured.

# Weaknesses
1. The proposed approach relies mainly on existing technology.
1. The theoretical contribution to robot learning is limited.
1. The experimental details, such as datasets, are insufficiently explained.
1. Comprehensive analysis and comparisons with conventional methods are insufficient.

### Post-rebuttal comment
The reviewers initially raised a few concerns. The authors provided a detailed response to each of these points. The reviewers agree that most of the concerns have been addressed. In their final scores, the reviewers are almost in agreement on the quality and acceptance. I support their shared opinion.